# Exploring the impact of a brief positive experience on dogs' performance and stress resilience during a learning task

Julia Miller[1,2]*, Camila Cavalli[1], Amin Azadian[1], Alexandra Protopopova[1]

1 Animal Welfare Program, Faculty of Land and Food Systems, The University of British Columbia, Vancouver, Canada, 2 Department of Immunology, Pathophysiology and Veterinary Preventive Medicine, University of Environmental and Life Sciences, Wroclaw, Poland

☯ These authors contributed equally to this work.
* julia.miller@upwr.edu.pl

## Abstract

Learning and stress resilience can be influenced by recent experiences. Research has traditionally focused on the effects of negative situations and stressors on subsequent learning and stress resilience, while knowledge is limited regarding the effects of positive experiences. We aimed to examine the impact of a pre-session brief positive experience on dogs' learning and stress resilience. Pet dogs were quasi-randomly assigned to the experimental (n = 20) or control (n = 20) group, counterbalanced for age, sex, and breed clade. Experimental dogs received a session intended to provide a positive experience, which included a 15 min walk on a long leash, human interaction, exploration, playing, and olfactory-based foraging. Control dogs were kept on leash in an office without being allowed to explore nor interact with their owner or the experimenters for 15 min. After 60 s of habituation to the testing room, all dogs were taught to nose-touch the experimenter's hand. After the Learning phase, there was a 2 min Disruption phase, in which a remote-controlled car moved inside of a tub at a distance. Measures included the number of hand touches in each phase, the latency to return to the task, and general stress and affiliative behaviours. No differences were observed in the Learning phase. Surprisingly, experimental dogs exhibited higher stress levels than control dogs during the Disruption phase. These dogs also spent a significantly higher proportion of time in proximity to their owners, which could be interpreted as reassurance-seeking behaviour. Contrary to our predictions, exposure to a brief positive experience did not impact learning and, surprisingly, seemed to have made dogs more susceptible to stress during the Disruption phase. Several possible explanations are discussed, including the possibility of an unintentional induction of a negative emotional state by the termination of the positive experience, as well as differences in arousal, or habituation to the indoor environment.

**Data availability statement:** All relevant data are within the manuscript and its Supporting information file

**Funding:** AP - Natural Sciences and Engineering Research Council of Canada Discovery Grant (RGPIN-2021-02591). The funders had no role in study design, data collection and analysis, decision to publish, or preparation of the manuscript.

**Competing interests:** The authors have declared that no competing interests exist.

## Introduction

The domestication history of dogs has resulted in their adaptation to the human social environment, trainability, and the development of understanding of communicative cues given by humans [1–4]. Despite this adaptation, living in a human-dominated environment can be associated with many challenges, including frequent encounters with unfamiliar objects, sounds, people, and other animals.

A key goal in improving the welfare of companion dogs is to understand how to create behavioural interventions in order to support dogs' resilience to these stressors. The definition of 'resilience' is continuously evolving and is typically thought of in terms of human functioning [5]. However, in a non-human animal context, resilience, while still having many definitions, may be defined as being able to cope with an environmental stressor [reviewed in 6,7]. Therefore, in a behavioural context, resilience may be assessed by the ability of the animal to "bounce back" following an administration of a stressor [7].

A dog's response to environmental stressors depends on individual traits shaped by multiple factors, such as genetics, early socialisation, and prior experience [e.g., 8–13]. Moreover, as noted by dog training professionals, dogs may also experience the phenomenon of "trigger stacking", in which their resilience to a relatively mild stressor is reduced due to an accumulation of prior stressors [14,15].

A variety of tests have been implemented to assess canine stress, including exposure to various ambiguous or novel stimuli (e.g., a vacuum cleaner, a startling sound, or a remote-controlled toy car [16–20]), and social situations (e.g., being left alone, being ignored by the owner, being alone with a neutral or friendly stranger, or even encountering a stranger exhibiting threatening behaviour [19,21–25]). The evaluation of dogs' stress during these tasks includes behavioural assessments and measurement of physiological parameters such as cortisol or heart rate [16,19,20,25].

Due to the unique human-dog relationship and the variety of roles dogs play in our society (e.g., pet, assistance, therapy, guarding), another field of extensive research focuses on learning capacity and the factors influencing the outcomes of dogs' training. These factors include genetics, training history, early experiences, age, and source of acquisition [26–30]. Nevertheless, learning is also likely impaired by stress [for a review see: 31,32]. For instance, research has shown that shelter dogs tend to perform worse than pet dogs in a variety of learning tasks, which has been attributed to their reduced experience interacting with people, along with the substantial stress they endure as a result of these living conditions [33–36]. Other animal studies have experimentally demonstrated the influence of aversive stimuli on cognitive performance. For example, social isolation and unfamiliar environments negatively affected pig performance in a spatial memory test [37]. Similarly, moving to a new environment temporarily reduced dwarf goats' performance in a visual shape discrimination task [38]. In sheep, pre-treatment with threatening stimuli and presenting white noise during the task negatively affected their performance in a spatial maze task [39]. Interestingly, in a study evaluating the performance of both owners and their dogs in a spatial working memory task, dogs who were stressed by being separated from their owners showed better

performance [40]. Given the relatively scarce literature on this topic, more research is needed to increase our understanding of the effects of stress on different types of learning tasks in companion animals.

The above-mentioned studies are part of a larger body of research focused on how animals respond to situations expected to induce specific emotional states, usually related to fear and anxiety [41].

Given that a negative experience may hinder subsequent learning, it becomes relevant to examine if positive experiences, instead, may enhance learning. However, there has been less focus on the potential influence of positive experiences on animal learning [but see 42]. To date, evidence for the impact of a positive affective state on different aspects of learning comes mainly from human studies. However, the results of these studies are mixed. While some show a beneficial impact of a positive affective state on different aspects of cognition, such as creative problem-solving or knowledge transfer [e.g., 43,44], others show opposite or mixed effects [e.g., 45,46]. Ways of inducing affective states in human participants include a pre-task treatment phase, such as the presentation of comedy videos versus neutral videos or videos with disturbing content [e.g., 43,45], performing a self-referencing mood-induction procedure [47], or a self-induced mood manipulation [48].

Whereas administering pain, restraint, or isolation is easy to conceptualize as "negative" for an animal and these experiences are often included in studies on animal welfare, procedures that might induce a positive state in an animal are studied much less frequently. Providing enrichment, such as additional sensory stimulation and positive human interaction has been demonstrated to result in general benefits to many animals [for a review see 49,50] and olfactory stimulation was shown to positively impact the behaviour of kenneled dogs as was shown by increased exploration and reduced stress-related behaviours [51]. A test widely used for evaluating emotional states (more precisely, optimism and pessimism) is the judgment bias test [for a review, see 52]. Briefly, this test focuses on the animal's behaviour towards an ambiguous stimulus, for instance, a bowl placed halfway between two locations previously established as positive (always contains food) or negative (always presented empty). The dog's latency to approach the ambiguous location is considered an indicator of their positive (optimistic) or negative (pessimistic) affective state, as it would be expected for them to approach faster if they are expecting food in the ambiguous location. Duranton and Horowitz [53] found that engaging dogs in olfactory-based activity (i.e., "nosework") for two weeks resulted in a more optimistic approach compared to the pre-session baseline. Olfactory enrichment with essential oils also generated a more optimistic response in shelter dogs [54]. Positive human interaction increased optimistic approach in fearful dogs housed in an animal shelter (however, the same procedure did not result in changes in optimism in non-fearful dogs) [55]. The provision of a complex toy resulted in a slightly more optimistic approach in pet dogs, albeit the effect might not have been robust [56]. Nevertheless, the data on the induction of positive affective states and their subsequent impact on dogs' behaviour and learning remains limited.

This study aimed to evaluate the impact of a brief positive experience (including exploration, playing with a toy, and olfactory-based foraging) on dogs' learning and subsequent stress resilience in the presence of a potentially stressful stimulus. We hypothesised that a pre-session positive experience, compared to a neutral experience, would result in better performance in the learning task and increased stress resilience, evidenced by a shorter latency to return to the learning task after the disruption began.

## Materials and methods

### Ethical statement

All procedures were approved by the University of British Columbia Animal Care Committee (A22-0170). Owners consented to the participation of their dogs in this study. No human-related data were obtained nor analysed during the study. The individuals seen in Figs 1 and 2 have given written informed consent (as outlined in PLOS consent form) to publish these pictures.

### Subjects

Fifty-two dogs were recruited to the study. The inclusion criteria for the study included being comfortable in unfamiliar environments and around strangers, as reported by the owner, not being already familiar with the hand touch command,

having high food motivation as reported by the owner, having current core vaccinations, and being between six months and ten years of age. Twelve dogs (eight neutered males and four neutered females; mean age = 4.67 ± 2.67 years) were excluded from the study because of early termination of the experiment due to fearfulness (n = 7, four from the experimental group) or not meeting the learning criteria of the task (n = 5, three from the experimental group). Forty pet dogs (mean age = 4.42 ± 3.05 years) were quasi-randomly assigned to the experimental (n = 20) or control (n = 20) groups, counterbalanced for age, sex, and breed clade (following Parker et al. [57]). The experimental group (mean age = 4.12 ± 2.18 years) included six neutered females, one intact female, 12 neutered males, and one intact male. The control group (mean age = 4.75 ± 2.59) consisted of eight neutered females, 11 neutered males, and one intact male. See Table 1 for detailed information on the dogs recruited for the study.

**Table 1. Identification numbers (ID No), breed, age, sex, and neuter status of dogs recruited into the study.**

| Control group | | | | Experimental group | | | |
|---|---|---|---|---|---|---|---|
| ID No | Breed | Age (years) | Sex | ID No | Breed | Age (years) | Sex |
| C2 | Labrador Retriever | 5 | spayed female | E1 | Mixed breed | 4 | neutered male |
| C3 | Mixed Breed | 2 | neutered male | E3 | Standard Poodle | 4 | neutered male |
| C4 | Labrador Retriever | 8 | neutered male | E6 | Golden Retriever | 2 | spayed female |
| C5 | Bernese Mountain Dog | 0.5 | intact male | E7 | Mixed breed | 8.5 | neutered male |
| C6 | Vizsla | 7.5 | neutered male | E8 | Mixed breed | 5 | neutered male |
| C7 | Portuguese Water Dog | 4 | spayed female | E10 | Mixed Breed | 0.5 | neutered male |
| C8 | Toy Poodle | 2 | neutered male | E11 | Toy Poodle | 4 | neutered male |
| C9 | Welsh Corgi Pembroke | 6 | spayed female | E12 | American Bully | 8 | neutered male |
| C11 | Beagle | 5 | spayed female | E13 | Mixed breed | 7 | neutered male |
| C12 | Mixed Breed | 3 | neutered male | E15 | Dachshund | 7 | spayed female |
| C13 | Mixed Breed | 2 | neutered male | E16 | West Highland WhiteTerrier | 3 | neutered male |
| C14 | Golden Retriever | 4 | neutered male | E17 | Mixed breed | 6 | spayed female |
| C17 | Biewer Terrier | 2 | neutered male | E18 | Mixed breed | 2 | spayed female |
| C18 | Mixed Breed | 1.5 | spayed female | E19 | Flat Coated Retriever | 6 | spayed female |
| C19 | Mixed Breed | 1.5 | spayed female | E22 | Golden Retriever | 5 | neutered male |
| C20 | Mixed Breed | 3 | spayed female | E23 | Labrador Retriever | 0.75 | intact female |
| C22 | Mixed breed | 4 | neutered male | E24 | Welsh Corgi Pembroke | 1 | spayed female |
| C23 | Welsh Corgi Pembroke | 9 | neutered male | E25 | Border Collie | 2 | neutered male |
| C24 | Duck Tolling Retriever | 10 | neutered male | E26 | Bernese Mountain Dog | 3 | neutered male |
| C25 | Bernese Mountain Dog | 7 | spayed female | E27 | Wirehaired Pointing Griffon | 1 | intact male |

Owners of the dogs who reached the learning criteria completed the Training and Obedience, Fear and Anxiety, and Attachment and Attention-seeking subscales of the Canine Behavioral Assessment and Research Questionnaire (C-BARQ, Hsu & Serpell [58]).

## Experimental conditions

For the experimental condition, dogs received a 15-min structured walk in a quiet outdoor area with their owner and two experimenters. See Fig 1.

The walk consisted of 5 min of free exploration, 5 min of playing with a toy (tug toy or ball, depending on the dog's engagement with the toy), and 5 min of olfactory-based foraging (sniffing for treats [Zuke's Mini Naturals®] scattered in the grass). The handler repeatedly tried to engage the dog in play, while the experimenter, who would later conduct the learning task, did not encourage interaction (i.e., not talking to the dog nor petting them if approached). If the dogs were not interested in any of the available toys, they continued with free exploration instead. The activities were chosen, based on previous research that found that human social interaction, access to toys, and olfactory search induce 'optimism' in dogs [53,54]. To encourage olfactory search, and given that dogs benefit from food-stuffed toys [59] we added treats as one of the elements of the positive experience session. The sequence of the tasks remained consistent across dogs. Exploration occurred first to allow for habituation to the experimenter and olfactory foraging was last to minimise potential frustration from removing access to food.

Dogs were fitted with a 4 m long leash held by one of the experimenters (i.e., handler) to ensure that the leash remained loose to reduce any potential sensation of physical restriction. All authors have extensive experience working professionally with dogs and utilised their skills to reduce any potential discomfort and increase positive affect (through observing dog body language and adjusting movements accordingly).

For the control condition, dogs spent 15 min with their owner and the two experimenters in an office space with three chairs and a table. During this time, the experimenters held a general conversation with the owners, and the owners were given the C-BARQ questionnaire to complete. If the owners did not finish completing the questionnaire during this part of the study, they continued after the experiment (similarly, owners of the experimental group dogs completed the C-BARQ questionnaire after the experiment). The dogs were kept on leash without being allowed to explore the room, and all attempts to interact with the people were ignored. Dogs in this condition did not receive any treats. See Fig 2.

The experimenters were JM or CC for all dogs. The handlers were usually CC or AP, except on three occasions when the handler was AA.

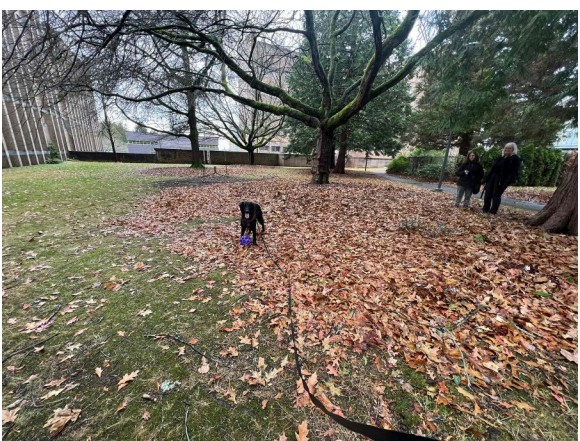

**Fig 1. Image of the setup in the experimental condition.** The structured walk consisted of 5 min of free exploration, 5 min of playing with a toy, and 5 min of olfactory-based foraging.

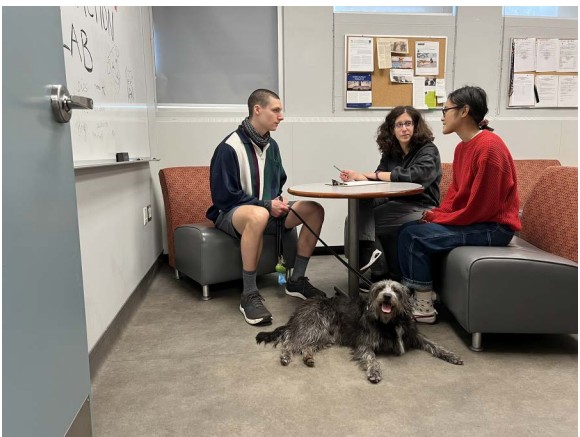

**Fig 2. Image of the setup in the control condition.** The dogs stayed with the owner and the experimenters in an office space.

## Task setup

The owner and the handler (or one of the experimenters from the control condition) sat in the corner of the room. The experimenter, responsible for teaching the hand-touch command, sat in the centre of the room (on a chair or the floor, depending on the dog's size), approx. 3.5 m away from the owner and the handler. A water bowl was located next to the wall opposite the tub, close to the owner. A plastic transparent tub (width 88 cm x depth 48 cm x height 32 cm) containing a remotely controlled car (LiteHawk REBEL®) was placed on the other side of the room, approximately 3.5 m away from the owner and the handler, and approximately 2 m away from the experimenter. A perimeter was marked on the floor at 30 cm from the tub and was later used to evaluate the proximity to the car during the analyses of the videos. See Fig 3.

The experimenter used her left or right hand to teach the hand touch command, depending on the owner's dominant hand. In the case of right-handed owners, the dogs were trained to touch the experimenter's left hand; in the case of left-handed owners (dogs' IDs: C4, E4), the dog was trained to touch the experimenter's right hand. Additionally, the tub containing the car was placed on the right or left side of the room to ensure each dog would touch the hand further from the tub. The handler counted the time with a stopwatch and gave the sign to switch on the remotely controlled car that would be used as a disruptor (see Procedure) but did not interact with the dog.

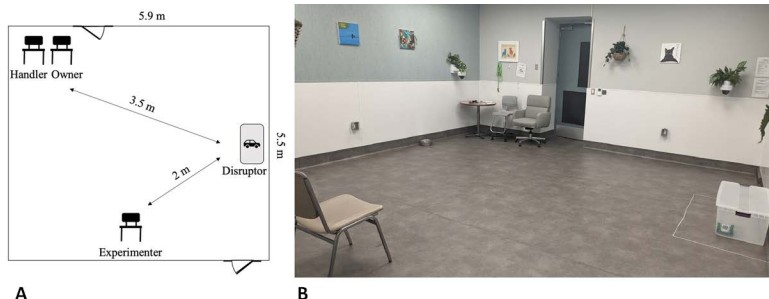

**Fig 3. Drawing (A) and image (B) of the setup during the task.** The handler and the owner sat in the corner of the room. The plastic tub containing a remotely controlled car was placed on the other side of the room. The experimenter sat in the centre of the room, approx. 3.5 m away from the owner and the handler.

All people were wearing dark sunglasses to avoid unintentional cueing caused by human gazing. The situation was filmed using a closed video circuit comprising four video cameras placed in each corner of the room.

The rewards for the task were small pieces of MaplelodgeFarms® chicken hot dog (each hot dog was sliced approximately 0.5 cm thick, and then each slice was cut into quarters to get several small treats). The rewards for six experimental (E1, E3, E6, E11, E16, E24) and seven control dogs (C2, C3, C4, C5, C8, C18, C23) were either Zuke's Mini Naturals® pieces in salmon flavour or other owner-provided commercial treats due to allergy concerns.

## Procedure

After the pre-treatment, all dogs were brought into the testing room and received a 60 s habituation phase. During this time, dogs were let off-leash and were free to explore the room while the experimenters explained the next phase of the task to the owners.

**Learning phase.** Immediately afterwards, the experimenter called the dog's name and fully extended her arm to the side, presenting the palm of her hand at the height of the dog's head. If the dog touched the hand (with any part of the body), the experimenter marked the behaviour with a verbal "yes" and tossed a treat on the floor in front of her. After the dog gazed back toward the experimenter, a new trial began, and the hand was presented again.

The learning criteria were met if the dog performed six subsequent touches with a latency of less than 3 s from the presentation of the hand to the touch. If the dog did not touch the hand within 10 s, the trial was considered a "no-choice", and the experimenter withdrew her hand and presented it again. After two consecutive no-choices, the experimenter presented her hand while holding a treat between her fingers to encourage the dog to approach it (i.e., assisted trial). Once the learning criteria were reached, the handler gave a sign (thumbs up) to a helper waiting in the control room to turn on the remotely controlled car in the plastic tub.

**Disruption phase.** In the Disruption phase, the experimenter continued to present her hand to the dog, while the car was switched on for 2 minutes. The helper was instructed to move the car erratically within the tub but refrain from constantly banging on the walls. The experimenter presented her hand immediately after the car was switched on and kept her arm extended continuously until the dog touched her hand for the first time. If that happened, the behaviour was again marked with a "yes" and rewarded with a treat, and the task continued as described in the Learning phase. In one case (dog's ID: E16), the car was switched off after 55 s because of excessive stress, as was evidenced by the body language and vocalisations (barking) of the dog.

**Food motivation control.** After the car was switched off, the experimenter waited 30 s without interacting with the dog. After this time, she tossed ten treats on the floor in front of her to evaluate food motivation at the end of the experiment. All dogs ate all of the treats during this phase. The purpose of this phase was to ensure that the dogs' engagement (or lack of engagement) in the task was not predominantly impacted by satiety.

## Data analysis

For the Learning phase, the "number of hand touches needed for acquisition", "number of no-choices", "number of assisted trials", and "latency to acquire the behaviour" were analysed. In the case of two dogs from the experimental group (E1, E18), the Learning phase included a short break in which the dog could freely explore the room again, as the dogs initially showed no interest in the interaction with the experimenter. For these two dogs, we have not analysed "latency to acquire the behaviour". For the Disruption phase, the "latency to return to the task" (first hand touch after switching on the car) and "number of hand touches" were analysed. Additionally, for both phases, videos were coded for affiliative behaviours toward the owner and the handler, body language related to stress, barking, and other behaviours (proximity to the door, sitting, lying down). For the Disruption phase, the videos were additionally coded for behaviours toward the stressor, and affiliative behaviours toward the experimenter. Note that affiliative behaviours toward the experimenter were not coded in the Learning phase because it was impossible to distinguish the dog's motivation, e.g., for

gazing at the experimenter at the time when they did not know the task and were engaged in learning. The coding was conducted using a partial-interval procedure with 5 s time bins. Occurrence or non-occurrence in each 5 s time bin was noted for each behaviour to accommodate a balance between recording event and state behaviours and allowing for the comparison across behaviours [60]. Due to a technical problem, sound was not correctly recorded for 12 dogs. For this reason, instead of coding all vocalisations, we decided to analyse only barking, as this behaviour can also be registered based on the dog's facial movements. See Table 2 for a full description of the analysed behaviours. The results are expressed as a proportion of the time bins in which each behaviour occurred. Because of the treat types used (soft pieces of chicken hot dogs) the lip-licking behaviour was counted as linked to stress only if it occurred in a time bin without food delivery. Single elements of body language were subsequently presented together as an "average stress score/time bin" (by summing up the number of unique behaviours that were scored as present in a given time bin and thus presenting the average stress 'score' per time bin).

**Table 2. Operational definitions of the behaviours that were observed during the Learning and Disruption phase.**

| Behaviour | Operational definition |
| --- | --- |
| **Affiliative behaviours** | |
| **Owner/handler proximity** | At least one paw placed within arm's length from the owner and/or handler |
| **Experimenter proximity** | At least one paw placed within arm's length from the experimenter while the dog is not participating in the task |
| **Gazing at owner/handler** | The eyes of the dog are directed at/in the direction of the owner and/or handler |
| **Gazing at experimenter** | The eyes of the dog are directed at/in the direction of the experimenter while the dog is not participating in the task |
| **Physical contact owner** | Any form of body contact with the owner initiated by the dog, including placing paws, leaning, or sniffing |
| **Physical contact handler** | Any form of body contact with the handler initiated by the dog, including placing paws, leaning, or sniffing |
| **Physical contact experimenter** | Any form of body contact with the experimenter initiated by the dog, including placing paws, leaning, or sniffing while the dog is not participating in the task |
| **Behaviours related to the disruptor** | |
| **Approaching the car** | Distance between the dog and the car is decreasing while the dog is not participating in the task |
| **Retreating from the car** | Distance between the dog and the car is increasing while the dog is not participating in the task |
| **Startle** | A sudden, sharp movement of the whole body |
| **Car proximity** | At least one paw or the head is within or on the tape measuring out 30 cm from the box with the car |
| **Gazing at the car** | The eyes of the dog are directed at the car |
| **Other behaviours** | |
| **Barking** | Opening the mouth with a rapid, rhythmic burst of movement |
| **Sitting** | The body is supported by two extended front legs and two flexed back legs |
| **Lying down** | The dog is lying down with limbs either tucked under or placed in front of the body |
| **Door proximity** | Any part of the dog is at arm's length or closer away from the door, the body oriented toward the door, including physical contact with the door (scratching, jumping, sniffing) |
| **Food delivery** | The dog is eating a treat |
| **Body language linked to stress** | |
| **Yawning** | Opening the mouth wide for at least 1 s |
| **Panting** | Breathing rapidly through opened mouth |
| **Lip licking** | Licking over lips or nose |
| **Shaking off** | Movements of body and/or head back and forth repeatedly and rapidly |
| **Paw lift** | Any paw is lifted off the ground for 1 s or more |
| **Tail down** | The tail is held between or through the hind legs or is forcibly held down (the base of the tail pressed to the body) |
| **Cowering** | The body is in a lowered, crouched position |
| **Hiding** | Hiding under a chair or table |

One observer coded all videos, while a second observer coded 30% of randomly selected videos. For most behaviours, the inter-observer agreement was high (range: 95.1% – 100%, mean: 97.4% ± 3.2%, min: 95.1% for "gazing at the car", and max: 100% for "cowering", "yawning", "shaking off" and "lying down"). The highest discrepancies were seen in coding "tail down" (58.3% − 100%), therefore this behaviour was double-coded in all of the videos from the Disruption phase with a mean agreement of 96.2% ± 8.1%.

To make sure that no single behaviour was over-interpreted as indicative of stress during the ethogram-based coding, both observers also scored the dogs' stress levels during the Disruption phase on a holistic 4-point scale (0 – no stress, 1 – low stress, 2 – moderate stress, 3 – high stress) based on their overall body language, barking, and other behaviours (i.e., seeking the proximity of the owner, and behaviour towards the object; see Table 3). The inter-observer agreement for this analysis was near perfect (Cohen's kappa = .96).

The stress level in the Disruption phase could be influenced not only by our experimental condition but also by the Learning phase, which took place directly before the Disruption phase. Therefore, a correlation was calculated to see whether there was a relationship between the "number of touches needed to acquisition" or "latency to acquire the behaviour" (which could contribute, for example, to longer habituation to the room or, on the contrary, frustration by lower success rate during the Learning phase) and the stress measurements of the Disruption phase.

**Statistical analyses.** Statistical analyses were conducted in Statistica 13 (StatSoft Tulsa, USA). A T-student test was used for normally distributed data (CBAR-Q Fear subscale; CBAR-Q Attachment subscale, "number of hand touches" in the Disruption phase). Non-parametric analyses (Mann–Whitney $U$ test to compare the two groups and Wilcoxon signed-rank test to compare the results in the Learning phase vs the Disruption phase within one group) were used for the rest of the data, that were not normally distributed ($p < .05$ in the Shapiro-Wilk test). The differences were considered significant if $p < .05$. Spearman's Rank correlation coefficient was used to analyse the relationship between the learning-related measurements in the Learning phase and the Disruption phase, as well as to analyse the relationship between the results of stress evaluation based on behavioural coding ("average stress score/time bin") and "overall stress level". Cohen's kappa was used to analyse the inter-observer agreement in the overall stress level assessment. The inter-observer agreement for individual behaviours was calculated by summing all agreements of whether or not a behaviour occurred in that interval, dividing it by the number of intervals, and multiplying by 100. Fig 4 was created with GraphPad Prism 10.1.2. Raw data used for the analyses are presented in S1 Dataset.

## Results

### C-BARQ scores

The results of the C-BARQ questionnaires are presented in Table 4. There were no statistical differences between the groups.

Table 3. Definitions used in the overall holistic stress assessment scale.

| Score | Definition |
|---|---|
| 0 (no stress) | The dog shows no signs of stress, or there is only a brief, mild reaction after turning on the car; the dog goes back to the task or lies down in the room; if looking at the car, the dog shows no signs of stress. |
| 1 (low stress) | The dog shows mild signs of stress after turning on the car; the reaction lasts longer than a few seconds or occurs repeatedly during the whole phase; for the majority of the phase, the dog engages in the task or seems to be rather distracted by the car than scared of it. |
| 2 (moderate stress) | The dog shows stress signs during the majority of the phase or intense stress signs after the car was turned on; even if engaging in the task for the majority of the phase, the dog is cautious and tense; the dog may repeatedly look at the car showing stress signs and/or look for reassurance in the proximity of the owner. |
| 3 (high stress) | The dog shows intense stress signs and/or is continuously looking for reassurance in the proximity of the owner while exhibiting stress signs; the dog touches the experimenter's hand only a few times or does not engage in the task at all. |

**Table 4. Results of the C-BARQ subscales.**

| | Fear & Anxiety | | Non-social Fear | | Training & Obedience | | Attachment | |
|---|---|---|---|---|---|---|---|---|
| | Mean ±SD | Student's t test | Median (IQR) | MWU test | Median (IQR) | MWU test | Mean ±SD | Student's t test |
| **Control** | 4.03 ±2.38 | $t=0.74$ $df=38$ $p=.462$ | 0.67 (1.00) | $U=149.50$ $p=.174$ | 2.56 (0.56) | $U=165.50$ $p=.355$ | 2.44 ±0.70 | $t=0.70$ $df=38$ $p=.290$ |
| **Experimental** | 3.49 ±2.27 | | 1.00 (1.25) | | 2.81 (0.63) | | 2.21 ±0.67 | |

Student's t test was used for normally distributed data; Mann Whitney U (MWU) test was used for data that were not normally distributed. No statistical differences were noted between the groups.

## Learning phase

No statistical differences were found between the control and the experimental group in the "number of hand touches needed to acquisition", "number of no choices", "number of assisted trials", nor "latency to acquire the behaviour". Similarly, no differences were significant in other parameters, including the "average stress score/time bin", affiliative behaviours, as well as "sitting", "lying down", and "barking". The proportion of time bins where the dogs stayed in the door proximity was significantly higher in the experimental group. See Table 5 for detailed results.

**Table 5. Stress-related, affiliative and task-related parameters, as well as other behaviours measured during the Learning phase.**

| | | Control | Experimental | Statistics |
|---|---|---|---|---|
| | | Median (IQR) | Median (IQR) | MWU test |
| **Stress-related parameters** | Average stress score/time bin | 0.19 (0.28) | 0.28 (0.91) | $U=156.50$ $p=.242$ |
| **Affiliative behaviours** | Owner/handler proximity | 0.00 (4.43) | 5.41 (8.93) | $U=143.00$ $p=.127$ |
| | Gazing at owner/handler | 7.39 (13.90) | 16.64 (14.73) | $U=142.00$ $p=.121$ |
| | Physical contact owner | 0.00 (1.99) | 0.00 (1.57) | $U=194.00$ $p=.883$ |
| | Physical contact handler | 0.00 (0.00) | 0.00 (1.57) | $U=185.00$ $p=.698$ |
| **Other behaviours** | Barking | 0.00 (0.00) | 0.00 (0.00) | $U=198.50$ $p=.968$ |
| | Sitting | 17.67 (37.06) | 2.94 (31.17) | $U=155.50$ $p=.231$ |
| | Lying down | 0.00 (0.00) | 0.00 (0.00) | $U=179.00$ $p=.583$ |
| | Door proximity | 0.00 (0.00) | 0.79 (4.23)[*] | $U=124.00$ $p=.040$ |
| **Task-related parameters** | Number of touches needed to acquisition | 10.00 (13.00) | 18.00 (8.50) | $U=139.00$ $p=.102$ |
| | Number of no-choices | 2.00 (4.50) | 4.00 (4.50) | $U=150.00$ $p=.183$ |
| | Number of assisted trials | 0.50 (1.00) | 1.00 (2.00) | $U=162.00$ $p=.314$ |
| | Latency to acquire the behaviour (s) | 102.00 (229.50) | 213.50 (137.00) | $U=135.00$ $p=.158$ |

[*]significantly different from control group at $p<.05$.

### Disruption phase

**Stress-associated body language and overall stress levels.** The "average stress score/time bin" was significantly higher in the experimental group than in the control group. The "overall stress level", as assessed by the observers, was also significantly higher than in the control group. See Fig 4. There was a significant positive relationship between the "overall stress level" and the "average stress score/time bin" ($r = .51$, $p = .002$).

**Task-related behaviours.** Dogs in the control group touched the hand more frequently than dogs in the experimental group. They also tended to return to the task faster than dogs in the experimental group. See Table 6.

**Affiliative behaviours.** Dogs in the experimental group spent a significantly higher proportion of time in the proximity of their owners than dogs in the control group. No significant differences were noted in the proportion of time bins in which other affiliative behaviours ("owner/handler proximity", "experimenter proximity", "gazing at owner/handler", "gazing at experimenter", "physical contact owner", "physical contact experimenter", "physical contact handler ") were observed. See Table 6.

**Other behaviours.** No significant differences were found in the proportion of time bins in which "barking" was observed. Similarly, there were no differences in other behaviours, like "startle", "lying down", "sitting", or "door proximity". Moreover, no differences were found in most of the behaviours linked to the disruptor ("gazing at the car", "approaching the car", "car proximity"). However, there was a tendency for a higher proportion of time bins where "retreating from the car" was observed in the experimental group versus the control group. See Table 6.

### Relationships between the learning phase and the disruption phase

The relationships between "latency to acquire the behaviour" and measurements of stress in the Disruption phase ("overall stress levels", and "average stress score/time bin") were not significant ($ps > .435$). Similarly, no relationships were found between the "number of touches needed to acquisition" and the stress measurements in the presence of the disruptor ($ps > .206$).

The "average stress score per time bin" was higher in the Disruption phase than in the Learning phase in both groups, however, the difference was statistically significant only for the experimental group. Dogs in the experimental group also barked more and spent a higher proportion of time in the proximity of their owners in the Disruption phase than in the Learning phase. No other differences were noted in the experimental group, and in the control group, none of the coded behaviours differed significantly between the two phases (Table 7).

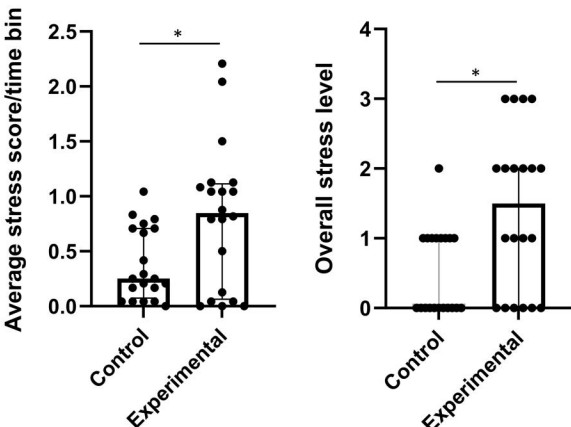

**Fig 4. Overall stress level and average stress score/time bin.** Median (box) and interquartile range (whiskers) of overall stress level and average stress score/time bin in the control and experimental group. Dots represent individual values. * $p < 0.05$.

**Table 6. Detailed results of the parameters measured during the Disruption phase.**

| | | Control | Experimental | Statistics |
|---|---|---|---|---|
| | | Median (IQR) | Median (IQR) | MWU test |
| **Stress-associated parameters** | Average stress score/per time bin | 0.25 (0.60) | 0.85 (1.02)* | U = 124.00 p = .040 |
| | Overall stress level | 0.00 (1.00) | 1.50 (2.00)* | U = 110.00 p = .014 |
| **Affiliative behaviours** | Owner proximity (%) | 0.0 (2.1) | 8.3 (25.0)* | U = 119.00 p = .028 |
| | Experimenter proximity (%) | 4.2 (22.9) | 0.0 (10.4) | U = 164.50 p = 0.341 |
| | Gazing at owner/handler (%) | 10.4 (14.6) | 16.7 (18.8) | U = 156.50 p = .242 |
| | Gazing at experimenter (%) | 4.2 (20.8) | 10.4 (22.9) | U = 194.00 p = .883 |
| | Physical contact owner (%) | 0.0 (0.0) | 0.0 (4.2) | U = 174.00 p = .495 |
| | Physical contact handler (%) | 0.0 (0.0) | 0.0 (0.0) | U = 199.50 p = .989 |
| | Physical contact experimenter (%) | 0.0 (2.1) | 0.0 (0.0) | U = 177.00 p = .547 |
| **Disruptor-directed behaviours** | Gazing at car (%) | 33.3 (37.5) | 58.3 (47.9) | U = 146.50 p = .127 |
| | Retreating car (%) | 8.3 (10.4) | 8.3 (18.8) | U = 130.50 p = .060 |
| | Approaching car (%) | 2.1 (8.3) | 6.3 (16.7) | U = 144.50 p = .134 |
| | Car proximity (%) | 0.00 (0.0) | 0.0 (0.0) | U = 191.50 p = .820 |
| | Door proximity (%) | 0.0 (0.0) | 0.0 (2.1) | U = 174.00 p = .495 |
| **Other behaviours** | Startle (%) | 0.0 (2.1) | 0.0 (4.2) | U = 184.00 p = .678 |
| | Barking (%) | 0.0 (0.0) | 0.0 (4.2) | U = 183.50 p = .659 |
| | Lying down (%) | 0.0 (0.0) | 0.0 (0.0) | U = 187.00 p = .779 |
| | Sitting (%) | 4.2 (37.5) | 0.0 (14.6) | U = 137.00 p = .091 |
| **Task-related behaviours** | Latency to return to the task (s) | 12.00 (27.00) | 23.50 (59.50) | U = 130.50 p = .060 |
| | On-task behaviours (%) | 79.2 (43.8) | 70.8 (70.8) | U = 146.50 p = .149 |
| | | Mean ±SD | Mean ±SD | Student's t test |
| | Number of touches | 12.10 (5.45) | 8.40 (5.89)* | t = 2.06 df = 38 p = .046 |

Student's t test was used for normally distributed data; Mann Whitney U (MWU) test was used for data that were not normally distributed.

*significantly different from control group at $p < .05$.

**Table 7. Detailed results of the comparison of the behaviours coded in the Learning phase and the Disruption phase.**

| | | Control | | | Experimental | | |
|---|---|---|---|---|---|---|---|
| | | Learning phase Median (IQR) | Disruption phase Median (IQR) | Wilcoxon signed-rank test | Learning phase Median (IQR) | Disruption phase Median (IQR) | Wilcoxon signed-rank test |
| **Stress-related parameters** | Average stress score/time bin | 0.19 (0.28) | 0.25 (0.60) | n = 19 Z = 1.83 p = .067 | 0.28 (0.91) | 0.85 (1.02)** | n = 19 Z = 3.06 p = .002 |
| **Affiliative behaviours** | Owner/handler proximity (%) | 0.00 (4.43) | 0.00 (2.08) | n = 9 Z = 0.77 p = .441 | 5.41 (8.93) | 8.33 (25.00)* | n = 15 Z = 2.10 p = .036 |
| | Gazing at owner/handler (%) | 7.39 (13.90) | 10.42 (14.58) | n = 19 Z = 1.57 p = .117 | 16.64 (14.73) | 16.67 (18.75) | n = 19 Z = 1.61 p = .107 |
| | Physical contact owner (%) | 0.00 (1.99) | 0.00 (0.00) | n = 7 Z = 1.01 p = .310 | 0.00 (1.57) | 0.00 (4.17) | n = 12 Z = 1.10 p = .272 |
| | Physical contact handler (%) | 0.00 (0.00) | 0.00 (0.00) | n = 8 Z = 0.70 p = .484 | 0.00 (0.00) | 0.00 (0.00) | n = 8 Z = 0.56 p = .575 |
| **Other behaviours** | Barking (%) | 0.00 (0.00) | 0.00 (0.00) | n = 5 Z = 0.67 p = .500 | 0.00 (0.00) | 0.00 (4.17)* | n = 6 Z = 1.99 p = .046 |
| | Sitting (%) | 17.67 (37.06) | 4.17 (37.50) | n = 15 Z = 0.91 p = .363 | 2.94 (31.17) | 0.00 (14.58) | n = 16 Z = 1.86 p = .063 |
| | Lying down (%) | 0.00 (0.00) | 0.00 (0.00) | n = 4 Z = 0.73 p = .465 | 0.00 (0.00) | 0.00 (0.00) | n = 5 Z = 0.67 p = .500 |
| | Door proximity (%) | 0.00 (0.00) | 0.00 (0.00) | n = 6 Z = 0.31 p = .753 | 0.79 (4.23) | 0.00 (2.08) | n = 10 Z = 0.25 p = .799 |

*significantly different from Learning phase at p < .05;

**significantly different from Learning phase at p < .01.

## Discussion

The goal of this study was to examine the effect of a brief positive experience on learning and stress resilience in dogs. Our results do not support our hypothesis that this experience would have a positive impact. Moreover, dogs exposed to a positive experience showed higher stress levels in the presence of the disruptor than those that did not receive the positive experience. They also spent a higher proportion of time in the proximity of their owners, which can be considered as looking for reassurance [e.g., 61–63]. In a study on the effect of a positive experience on affective state in laboratory Beagles, Burman and colleagues [64] also did not observe a positive effect of a relatively similar positive experience (for-aging for treats in a maze) on the outcome of a judgment bias test. Moreover, the dogs exposed to the experience showed more "pessimistic" responses than the control dogs [64]. This, in line with our findings, suggests that a brief session of positive experiences may not be sufficient to induce a sustained positive affective state in dogs and may even create a subsequent negative affective state.

There are several possible explanations for our results. Some of them highlight the possibility of our pre-treatment modulating stress susceptibility during the Disruption phase in unexpected ways. One such possibility is that experimental dogs may have been experiencing a higher level of arousal after the pre-treatment, which continued during the Learning

and Disruption phases. Arousal-Valence Models [41,65] comprise a framework which categorizes emotions based on the intersection between the dimensions of arousal (from high arousal to low arousal) and valence (positive to negative). As such, if the Disruption phase was interpreted negatively (contrary to the Learning phase, where stress scores were lower and did not differ between the groups) it would be possible for highly aroused individuals to express more fearful or anxious responses in the presence of a stressor.

Another possible explanation is an unintentional induction of a negative emotional state by the termination of the positive experience when moving to the laboratory, whereas dogs in the control condition might have been experiencing boredom and/or frustration caused by lack of attention and, therefore, experienced a shift towards a positively-valenced emotional state when the "neutral" experience was terminated [64,66]. A similar result was found in sheep, which showed more optimistic approaches in a cognitive bias test after being restrained [67]. In this study, sheep from the experimental group were subjected to a 6h restraint and isolation stress for three consecutive days and were tested daily in a judgement bias test immediately afterwards. Contrary to the initial hypothesis, sheep from the experimental group were more likely to approach the ambiguous locations compared to control sheep, who were not subjected to restraint and isolation. The authors suggested that the release from restraint immediately before testing could have resulted in a more positive emotional state. However, if this were true for the dogs in our study, we would expect to see differences in stress-related behaviours between the two groups of dogs in the first phase of the experiment (the Learning phase), but we did not see this difference.

Another alternative is that, although control dogs spent 15min in an office space and not in the laboratory where the Learning and Disruption phases were later conducted, we cannot exclude the impact of their prolonged exposure to the smells or sounds inside of the building on their subsequent stress levels in the presence of a novel stimulus. Thus, lower stress scores in the control dogs might be associated with their longer habituation to the indoor environment, and result in a lack of heightened stress scores in this group after the introduction of the stressor. This explanation could also be supported by the fact, that dogs from the experimental group spent significantly more time in the proximity to the door (which may reflect they unwillingness to stay in the room) during the Learning phase. In an experiment comparing the behaviour of dogs and cats during a habituation test in laboratory settings (first visit to the laboratory), the authors concluded that dogs generally do not require habituation before laboratory tests, as all the participating dogs successfully met the criteria for habituation [68]. However, it is important to note that the minimum length of the habituation phase in the cited study was 5 minutes -significantly longer than the 1-minute habituation period in our study design. This discrepancy suggests that our habituation period might have been insufficient for the experimental group, especially when compared to the potential habituation experienced by the control group.

Finally, another interesting possibility could be that the waiting may have been actually stressful for control dogs, and this prior stressful experience could have prepared them to deal better with the stressor during the Disruption phase. Exposure to moderately stressful events (i.e., "stress inoculation") has been identified as a factor promoting resilience to subsequent stressful episodes in several species [for a review, see, 69]. However, these studies usually focus on early life stressors, and more research is needed to further explore this idea. Anecdotally, it should be noted that we did not observe excessive stress signs (other than mild frustration and attention-seeking behaviours) in dogs in the control condition. However, the pre-session conditions (both the positive and the neutral experience) were not recorded, so it is not possible to formally examine the dogs' behaviour during these events.

Another option that should be considered has to do with the delay between the pre-treatment and the Disruption phase. In the present study, experimental dogs were exposed to a positive experience before proceeding to subsequent phases of the study (Learning phase and Disruption phase). Therefore, the phase in which stress signs were analysed was preceded by a 60s habituation phase as well as a learning task lasting 32–489s, depending on the number of trials the dog needed to pass the acquisition criteria. As such, it is possible that this delay may have been too long for our treatment to have a positive impact on the subsequent task or that the behaviour of dogs in the

Disruption phase was influenced more by the Learning phase than by the pre-treatment condition. However, this is unlikely, as we found no differences in the stress score between the groups in the Learning phase, nor a correlation between the number of trials or time needed to acquire the hand-touch behaviour and the stress scores in the Disruption phase.

Finally, we should take into account that the analysis of stress-associated behaviours in our study was based on video coding of single elements of body language (subsequently summed to a body language stress score), barking and behaviours directed toward the object, the owner, the handler and the experimenter. While the inter-observer agreement was high, there were discrepancies in the coding of body language, particularly the tail position. This is why we decided to add another measure, a holistic stress scale, which showed a moderate positive correlation with stress scores based on body language. This may have been a result of including other behaviours in the holistic assessment, like barking, retreating from the object or physical contact with the owner, as these behaviours were coded separately from the stress body language during behavioural coding. While the overall inter-rater agreement for the holistic scale was near perfect, we cannot exclude that the high agreement was a consequence of this scale being less accurate than behavioural coding. However, in some cases, dogs that achieved relatively high numbers in body language stress scores were evaluated by both observers as not stressed, which can be explained by their physical appearance (e.g., holding the tail down in a neutral position, a very short tail making the analysis of its position questionable) or panting caused by individual sensitivities to temperature or excitement associated with getting food rewards. Also, the tail position could not be assessed in the time bins in which the dog was sitting or lying down. Developing unbiased and reliable methods for evaluating behavioural responses is a big challenge in research [70,71]. Similarly, a validated observation tool for measuring affective states and welfare in dogs is still lacking [for a review, see 72]. This may point to the importance of using multiple tools (e.g., behavioural analysis and evaluation of physiological parameters) to assess dogs' behaviour and affective states more accurately [e.g., 16,73], or using less common methods that reduce the impact of inter-rater discrepancies and bias, like crowdsourcing [74]. Additionally, qualitative methods have been implemented in recent years to facilitate the assessment of dogs' welfare or to complete the results obtained from quantitative behavioural analyses [75,76]. Another novel approach is the use of automatic tracking and scoring of dogs' behaviour, which reduces the risk of observer bias [77–79]. Thus, we encourage further discussion and research in this area, promoting the development of more accurate methods for evaluating animals' emotional states.

In terms of learning effects, our results do not suggest an impact of the experimental condition on motivation and learning. It is possible that due to the characteristics of the control and experimental conditions, additional factors might have influenced the results in the subsequent tasks. These could include levels of satiety or the impact of exercise in the experimental condition. While both of these factors may be linked to perceived positive emotions, further research is needed to disentangle the possible contribution of each of these factors separately.

Limitations of our study include lack of data on the dogs' emotional state during and after being exposed to the experimental or control condition, including the level of their arousal. Based on the previously cited literature, we chose an experience that was hoped to be positive for dogs, but the procedure itself was not recorded. Our methodology could also have resulted in erroneous findings; many assumptions had to be made such as the valence of the brief experience, the aversive quality of the toy car, and that our measures represented the concept of 'resilience' – all of which could be debated.

Another limitation is the lack of detailed questionnaires about some dogs who did not reach the learning criteria. For these dogs, if the owner did not fill out the questionnaire during the experiment (e.g., due to limited time), we did not collect C-BARQ scores after the experiment. While it is common to report the number of animals that were excluded from the dog cognition studies or analyses as well as the reason for exclusion [e.g., 3,28,42], possible underlying factors are rarely analysed. Unfortunately, we did not collect additional data, such as the dogs' food motivation, that could help us analyse the factors underlying their failure.

## Conclusion

We found no positive impact of a brief positive experience on dogs' performance in a learning task and stress resilience in the presence of a stressful stimulus. On the contrary, dogs from the control group expressed fewer stress signs in the presence of the disruptor. The most parsimonious explanation for our results is that the control group may have had a longer habituation to the indoor environment. Other explanations may be a shift toward negatively valenced emotions after terminating the positive event in dogs from the experimental group, or a "stress inoculation" of dogs in the control group. More research is needed to evaluate the effects of affective states on dogs' behaviour during cognition tasks or in the presence of environmental challenges.

## Supporting information

**S1 Dataset. Raw data used for the statistical analyses.**
(XLSX)

## Acknowledgments

We would like to express our gratitude to our canine participants and their human companions. We would like to foremost thank Lucia Kotianova for her help with coding the videos. We are thankful to our colleagues in the Animal Welfare Program at the University of British Columbia, Bailey Eagan, Cheng Yu Hou, Katie Koralesky, Joseph Krahn, Lexis Ly, and David Schuman, who provided assistance during the study. We also thank the Dr. and Mrs. A. S. Dekaban Foundation for supporting the Polish-Canadian academic exchange.

## Author contributions

**Conceptualization:** Alexandra Protopopova.

**Formal analysis:** Julia Miller.

**Funding acquisition:** Alexandra Protopopova.

**Investigation:** Julia Miller, Camila Cavalli, Amin Azadian, Alexandra Protopopova.

**Methodology:** Julia Miller, Camila Cavalli, Amin Azadian, Alexandra Protopopova.

**Project administration:** Julia Miller, Camila Cavalli, Amin Azadian.

**Supervision:** Alexandra Protopopova.

**Visualization:** Julia Miller.

**Writing – original draft:** Julia Miller, Camila Cavalli.

**Writing – review & editing:** Julia Miller, Camila Cavalli, Amin Azadian, Alexandra Protopopova.

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
