## [Decision Letter · Decision Letter 0]

PONE-D-24-09989Exploring the impact of a prior positive experience on dogs' performance and stress resilience during a learning taskPLOS ONE

Dear Dr. Miller,

Thank you for submitting your manuscript to PLOS ONE. After careful consideration, we feel that it has merit but does not fully meet PLOS ONE’s publication criteria as it currently stands. Therefore, we invite you to submit a revised version of the manuscript that addresses the points raised during the review process.

We look forward to receiving your revised manuscript.

Kind regards,

I Anna S Olsson, Ph.D.

Academic Editor

PLOS ONE

Journal Requirements:

"AP - Natural Sciences and Engineering Research Council of Canada Discovery Grant (RGPIN-2021-02591)"

3. Please note that funding information should not appear in the Acknowledgments section or other areas of your manuscript. We will only publish funding information present in the Funding Statement section of the online submission form. Please remove any funding-related text from the manuscript. 

4. We note that Figures 1 and 2 includes an image of a participant in the study. 

As per the PLOS ONE policy (http://journals.plos.org/plosone/s/submission-guidelines#loc-human-subjects-research) on papers that include identifying, or potentially identifying, information, the individual(s) or parent(s)/guardian(s) must be informed of the terms of the PLOS open-access (CC-BY) license and provide specific permission for publication of these details under the terms of this license. 

Please download the Consent Form for Publication in a PLOS Journal (http://journals.plos.org/plosone/s/file?id=8ce6/plos-consent-form-english.pdf). 

The signed consent form should not be submitted with the manuscript, but should be securely filed in the individual's case notes. Please amend the methods section and ethics statement of the manuscript to explicitly state that the participant has provided consent for publication: “The individual in this manuscript has given written informed consent (as outlined in PLOS consent form) to publish these case details”. 

**Additional Editor Comments:**

Your paper has been reviewed by three independent reviewers, whose detailed comments are available at the end of this message. As you will see, there are some significant criticisms of the experimental design. Whether you will be able to address these convincingly will most likely be decisive for the outcome of a revision, should you decide to submit a revised version.

Reviewers' comments:

Reviewer's Responses to Questions

**Comments to the Author**

1. Is the manuscript technically sound, and do the data support the conclusions?

Reviewer #1: Partly

Reviewer #2: Yes

Reviewer #3: Partly

2. Has the statistical analysis been performed appropriately and rigorously? 

Reviewer #1: No

Reviewer #2: Yes

Reviewer #3: Yes

3. Have the authors made all data underlying the findings in their manuscript fully available?

Reviewer #1: Yes

Reviewer #2: Yes

Reviewer #3: Yes

4. Is the manuscript presented in an intelligible fashion and written in standard English?

Reviewer #1: Yes

Reviewer #2: Yes

Reviewer #3: Yes

5. Review Comments to the Author

Reviewer #1: In the study the authors aim to assess the effect of a positive experience in subsequent learning (i.e., nose touch the experimenter’s hand). Additionally, they introduced a post-learning disruptive, potentially stressful phase and recorded its effects on 2 groups of animals. The experimental group had a pre-experimental 15-min period of positive interactions, while the control group did not. The authors report no effects in learning across groups, but, against their predictions, the experimental subjects were more affected during the disruptive phase of the experiment.

MAIN POINTS:

Exploring the effect of positive experiences in learning and stress resilience is an interesting and relevant topic, but the current manuscript suffers from a series of weaknesses at multiple levels: experimental design (e.g., no within subject baseline, unbalanced data comparisons across experimental conditions, the control condition does not seem the most appropriate), data analyses (e.g., coarse sampling of behavioural data, only a subset of these metrics are used when comparing the different experimental phases, seemingly post-hoc analysis, i.e., holistic score).

Moreover, even though the authors are very thorough in detailing descriptive statistics, there is a general lack of information on all statistical tests (other than p values). Still in the statistics front, it is unclear what data were normal, and when this was not the case, why didn’t the authors try to transform it (e.g., log transform in case of latencies) before considering non-parametric alternatives.

Overall, the Introduction could use some streamlining, paragraphs read somewhat disjointedly. With regards to the Discussion, similarly to the Introduction, it lacks a clear train of thought, while it could also be more balanced in terms of discussing the data presented versus making speculative considerations that said data cannot help elucidate.

DETAILED POINTS

Line 38. reference to recent ManyDogs pointing paper would be a good addition (https://doi.org/10.26451/abc.10.03.03.2023)

Lines 46, 54, 70, 75. paragraphs would be justified here

Line 83. What do you mean by “Animal’s interpretation”? This is a very loaded word, consider rephrasing.

Line 85. “Contains” consider “Containing”

Line 99. “single positive experience (including exploration, playing with a toy, and olfactory-based foraging)”

Is this really a single experience, or multiple? If the authors have reasons to support the former this should be alluded to.

Also, the stress resilience comes a bit as a surprise, maybe it would be worth adding some words beforehand, to introduce the topic to a less expert reader.

Line 111. Isn’t it simpler to say you have recruited 52 and excluded 12?

Lines 120-130. It would be nice, and easier for the reader, to have subject data summarized on a table. This would also lighten the reading of the paragraph.

Starting on Line 135. Consider re-structuring: after introducing the walking, segue to what the walking consisted of, and end with other details.

Line 139. “did not encourage interactions”. It is unclear if this was done actively or in a passive manner. Also, it would be good to have a more detailed reasoning for the choice, duration and order of the activities used.

Lines 154-168. Consider re-structuring this paragraph. (e.g.,The setup should have been described before telling the read what it was used for.)

Line 162. There might be an issue with Figure 2, it does not show the task setup, it seems to be an example of the structured walk experimental animals were exposed to. Are we missing something here? Figure 1 is also never referred to in the main text.

Line 169. We can see where the authors are coming from, but, by no means, sunglasses might prevent dogs from seeing people’s eyes, but they don’t prevent any unintentional cueing. Please either elaborate or tone down this sentence.

Line 173. “To increase the dogs’ motivation for this part of the task”. What do the authors mean? Isn’t the point of the study to conduct a simple reward-based learning task. The authors even say they only recruit animals that are high food motivated. Consider removing.

Line 175. This information can be added to the Table with the subject information.

Line 198. Can the authors be more concrete? Why was this dog excessive while all others were not, and who was this subject? If the authors what to highlight subsets of animals, then let the reader know which ones.

Line 198. “by body language and vocalizations”. Consider adding “the” before “body language”.

Lines 199-203. Despite saying this phase was similar to the Learning Phase it is unclear if the experimenters kept doing trials for the duration of the Disruption.

Line 204. This seems a side point, it is unclear what is the purpose of this. How is this a food motivation control for the disruption?

Line 209. How were these metrics extracted, live by the helper, or from coding recorded videos.

Line 211. As stated before, dog IDs would be valuable information for the more curious reader.

Line 213. Who is the Experimenter 1, before authors refer to experimenter and helper. Are we missing something? Please be consistent with the terminology used.

Line 216. “for this phase, videos were coded for affiliative behaviours”. For the sake of comparison, and since we believe the video data are available, why not do this for the previous learning phase as well.

Line 218. “The coding was conducted using a partial-interval procedure with 5 s time bins. Occurrence or non-occurrence in each 5 s time bin was noted for each behaviour”. Can the authors justify this approach? Even if the authors end up binning their metrics, it is unclear why downsampling it from the start is the best way to proceed. Also, why did the authors not consider using the duration of these events instead.

Line 224. As previously stated, ID of the animal is useful information.

Line 226. “The lip-licking behaviour was counted as linked to stress only if it occurred in a time bin without food delivery”. Why is this the case, is it not possible that animals are stressed during food delivery? Please elaborate.

Line 228. “by summing up the number of stress signs in each time bin and presenting the average stress score per time bin”. Once again, the authors approach is to reduce data granularity even further without a clear justification for it. Along the same lines, what does something like “presence of stress” add to a finer measurement like “average stress score per time unit”? They are highly correlated (as can be seen from Figure 3. The latter adds no information to the former. Please consider revising the data presented in the figure as well.

Line 231. Table 1. Once again, the use of experimenter 1 and 2 is not consistent, early on the manuscript they were referred to as helper and experimenter. “Proximity to the car”. The definition is not clear, plus a measuring tape was never mentioned before (or after). More generally, if these descriptions were used before, a citation would be justified, if they have not, some of the description of the target behaviours should be expanded.

Line 237. What exactly is the statistic used to assess Inter-observer reliability? Please elaborate.

Line 242. “over-interpreted as indicative of stress”. This holistic analysis can also do exactly the opposite. All definitions (Table 2) seem very post-hoc, and the lack of granularity in the measurement alone can explain the high degree of inter-observer agreement. It is unclear what this adds the analyses and interpretation of the results.

Line 253. Unclear what the correlation was used for. Consider re-phrasing. and please add details.

Line 262. What were the statistics for the C-BARQ scores? t-tests? more information is needed here (e.g., t-statistics, df…). ps? How many and what tests were conducted? Were there any type of corrections for multiple comparisons? These results fall a bit short, they are not mentioned again in the manuscript.

Line 293. Please refer to the previous comment with regards to the stress measures used.

Line 298. “Dogs in the experimental group spent a significantly higher (U = 119.00, p = .028) proportion of time in the proximity of their owners”. As mentioned before, it would be interesting to see this data in comparison to the Learning phase.

Line 315. “Relationships between the Learning phase and the Disruption phase”. As previously stated, if this comparison is relevant, why only looking at a subset of metrics available?

Line 323. “Our results show that this experience had no positive impact.” At least with regard to learning, absence of evidence is not evidence of absence. Please consider rephrasing.

Line 331. Without a baseline metric for each dog, this is hard do assess. Moreover, in the 15 min pre-test phase, dogs are exposed to multiple things…it is unclear why this would be akin to a single rewarding event. If one uses the authors’ analyses criterium of 5 s bins, 15 minutes would be divisible into 180 units, each potentially associated with rewarding events such as walking, playing, olfactory exploration.

Line 338. Consider referring to these in the Introduction as well. Yet, as it stands it is unclear how this topic can help interpreting the results, as there are no data available that would allow for it (e.g., pre-experimental baseline data).

Line 343-347. Please see general comment regarding the balance between data driven and speculative remarks.

Line 349. Replace “can not” by “cannot”.

Line 365. “However, the pre-session conditions (both the positive and the neutral experience) were not recorded, so it is not possible to formally examine the dogs’ behaviour during these events.” This information is key to interpret the results obtained, a major flaw of the study. Please see general comment regarding experimental design.

Line 409. “However, we have found a significant difference in the learning performance between dogs given commercial treats and hot dogs”. Given that the experiment was not designed to test this, this is pure speculation. Maybe this is related to other features of particular dogs, that might have caused said differences. As acknowledged by the authors, no preference tests were conducted, and there are no data regarding the animals diet, for example (Line 424).

Reviewer #2: The study investigated the impact of immediate positive experience on dogs' performance and stress resilience during a learning task. In general, the manuscript is well written with very clear methodological details and interpretation and discussion of results.

I have a few minor suggestions that can improve the clarity of the manuscript more. In particular, 1) the literature cited in the current manuscript do not cover different subpopulations of dogs, so, consider exploring the work by Range and Bhadra groups on free-ranging dogs for more recent discoveries in the field of canine science. 2) Why authors did not use a randomised sequence of the learning and disruption phase? 3) Was the amount of 'stress' (by moving a toy car) enough to elicit stress-related behaviours, which authors also associated with 'negative emotional arousal with negative valence'. 4) The frequent use of 'affective states' while affect was actually not measured. Note, stress is not synonymous with affective states (like emotions and moods). I would suggest the authors to carefully review the sections where they try to explain stress with affective states. 5) While writing two different paragraphs, the second one should be a direct continuation of the first. So, please avoid starting paragraphs with 'similarly', 'These studies', 'In addition to', etc. 6) 'A positive prior experience' sounds vague, what about replacing with 'brief'. As the positive experience period included walking on a long leash, playing with toys, etc. Practically, you cannot tease apart the effects of those events from each other (as you haven't measured affective states).

Line 38:'human-made' sounds weird, please replace with 'human-dominated'.

Line 41-42: The current study did not explore genetic background associated with the behavioural responses. Therefore, I would suggest adding more relevant literature, see e.g.,

Bhattacharjee, D., Sarkar, R., Sau, S., & Bhadra, A. (2021). Sociability of Indian free-ranging dogs (Canis lupus familiaris) varies with human movement in urban areas. Journal of Comparative Psychology, 135(1), 89.

Brubaker, L., & Udell, M. A. (2018). The effects of past training, experience, and human behaviour on a dog’s persistence at an independent task. Applied Animal Behaviour Science, 204, 101-107.

Line 45-46: see the study below that investigated how threatening may shape behaviour in dogs -

Stellato, A. C., Flint, H. E., Widowski, T. M., Serpell, J. A., & Niel, L. (2017). Assessment of fear-related behaviours displayed by companion dogs (Canis familiaris) in response to social and non-social stimuli. Applied Animal Behaviour Science, 188, 84-90.

Bhattacharjee, D., Sau, S., & Bhadra, A. (2018). Free-ranging dogs understand human intentions and adjust their behavioral responses accordingly. Frontiers in Ecology and Evolution, 6, 232.

Also see, Lazzaroni, M., Schär, J., Baxter, E., Gratalon, J., Range, F., Marshall-Pescini, S., & Dale, R. (2023). Village dogs match pet dogs in reading human facial expressions. PeerJ, 11, e15601.

Line 81: A 'cognitive' test..

Line 331-333: See my comment on pt. 6.

Reviewer #3: Please see the attached file for my detailed comments.

The paper needs more clarification for better understanding of the methodology. The statistics are clear, and the language is fine. The results add to the growing understanding of dog-human interactions.

6. PLOS authors have the option to publish the peer review history of their article (what does this mean? ). If published, this will include your full peer review and any attached files.

**Do you want your identity to be public for this peer review?** For information about this choice, including consent withdrawal, please see our Privacy Policy .

Reviewer #1: No

Reviewer #2: No

Reviewer #3: **Yes: ** Anindita Bhadra

---

## [Author Response · Author response to Decision Letter 1]

26 Dec 2024

We would like to thank the reviewers and editor for the constructive comments we have used to refine our manuscript, including additional video coding, statistical analyses as well as extensive re-writing of the introduction and discussion section.

Below, we present our point-to-point answers to the reviewers’ and editor’s comments.

EDITOR:

Authors: We have made appropriate changes in the file information and headings.

"AP - Natural Sciences and Engineering Research Council of Canada Discovery Grant (RGPIN-2021-02591)"

A: We have added the information in the cover letter.

3. Please note that funding information should not appear in the Acknowledgments section or other areas of your manuscript. We will only publish funding information present in the Funding Statement section of the online submission form. Please remove any funding-related text from the manuscript.

A: We have removed the information on the funding from this section. We kept acknowledgements for the foundation that supported the academic exchange, as the study itself was not founded by the foundation.

4. We note that Figures 1 and 2 includes an image of a participant in the study.

A: The pictures are showing the setup from pilot trials, and as such are showing team members and not dog owners. We did obtain written consent to publish their images, and state it in the ethical statement, and separately for each figure (lines 120, 145, 172).

5. Please include captions for your Supporting Information files at the end of your manuscript, and update any in-text citations to match accordingly. Please see our Supporting Information guidelines for more information.

A: The caption is included at the end of the manuscript.

Reviewer #1:

MAIN POINTS:

Exploring the effect of positive experiences in learning and stress resilience is an interesting and relevant topic, but the current manuscript suffers from a series of weaknesses at multiple levels: experimental design (e.g., no within subject baseline, unbalanced data comparisons across experimental conditions, the control condition does not seem the most appropriate), data analyses (e.g., coarse sampling of behavioural data, only a subset of these metrics are used when comparing the different experimental phases, seemingly post-hoc analysis, i.e., holistic score).

Authors: Thank you for the thorough review of our study. We have made substantial changes to the manuscript, which include additional analyses (behavioural coding of videos from the learning phase as suggested, which was followed by additional data analyses – see detailed comments below. For comments on weaknesses we could not change at this stage (e.g. not having data on the dogs' emotional states during the experimental and control condition – since we have not recorded these events) we have added specific comments in the manuscript (see detailed comments below).

Moreover, even though the authors are very thorough in detailing descriptive statistics, there is a general lack of information on all statistical tests (other than p values). Still in the statistics front, it is unclear what data were normal, and when this was not the case, why didn’t the authors try to transform it (e.g., log transform in case of latencies) before considering non-parametric alternatives.

A: We have added the missing information on the data distribution (line 295) . The log transformation of the data (latency to acquisition, no of touches to acquisition, C-BARQ Non-social Fear subscale, C-BARQ Trainability subscale) did not result in normal data distribution; we have not decided to log transform other data because of the need for further data manipulation for data sets with “0” being a common value.

Overall, the Introduction could use some streamlining, paragraphs read somewhat disjointedly. With regards to the Discussion, similarly to the Introduction, it lacks a clear train of thought, while it could also be more balanced in terms of discussing the data presented versus making speculative considerations that said data cannot help elucidate.

A: We have re-written the introduction and discussion, including the removal of some points from the discussion, e.g. on the impact of different treats used in the learning task.

DETAILED POINTS

Line 38. reference to recent ManyDogs pointing paper would be a good addition (https://doi.org/10.26451/abc.10.03.03.2023)

A: we have added the citation (line 39, Reference 1)

Lines 46, 54, 70, 75. paragraphs would be justified here

A: The introduction was re-written taking into account all the reviews.

Line 83. What do you mean by “Animal’s interpretation”? This is a very loaded word, consider rephrasing

Line 85. “Contains” consider “Containing”.

A: Both these phrases were removed as we decided to remove a detailed explanation of the judgement bias test.

Line 99. “single positive experience (including exploration, playing with a toy, and olfactory-based foraging)”

Is this really a single experience, or multiple? If the authors have reasons to support the former this should be alluded to.

Also, the stress resilience comes a bit as a surprise, maybe it would be worth adding some words beforehand, to introduce the topic to a less expert reader.

A: We have changed “single positive experience” to “brief positive experience” throughout the manuscript, including the title.

We have also added the concept of stress-resilience in the Introduction (line 43).

Line 111. Isn’t it simpler to say you have recruited 52 and excluded 12?

A: We have re-written the paragraph (line 125)

Lines 120-130. It would be nice, and easier for the reader, to have subject data summarized on a table. This would also lighten the reading of the paragraph.

A: We have added a table summarizing the basic subject data (Table 1)

Starting on Line 135. Consider re-structuring: after introducing the walking, segue to what the walking consisted of, and end with other details.

A: The paragraph was re-structured as requested (line 144).

Line 139. “did not encourage interactions”. It is unclear if this was done actively or in a passive manner. Also, it would be good to have a more detailed reasoning for the choice, duration and order of the activities used.

A: We have added more information on the behaviour of the experimenter (line 155), as well as on the choice, duration and order of the activities (line 157)

Lines 154-168. Consider re-structuring this paragraph. (e.g.,The setup should have been described before telling the read what it was used for.)

A: We have re-written the paragraph (line 179), starting with the setup (leaving some information on the tasks the researchers were responsible for as it seems easier to picture this way).

Line 162. There might be an issue with Figure 2, it does not show the task setup, it seems to be an example of the structured walk experimental animals were exposed to. Are we missing something here? Figure 1 is also never referred to in the main text.

A: Thank you for capturing this obvious mistake. We have added a figure of the setup (Fig 1 now shows the experimental condition, Fig 2 the control condition and Fig 3 a drawing and corresponding picture of the setup)

Line 169. We can see where the authors are coming from, but, by no means, sunglasses might prevent dogs from seeing people’s eyes, but they don’t prevent any unintentional cueing. Please either elaborate or tone down this sentence.

A: The sentence was rephrased to: “All people were wearing dark sunglasses to prevent unintentional cueing caused by human gazing” (line 201).

Line 173. “To increase the dogs’ motivation for this part of the task”. What do the authors mean? Isn’t the point of the study to conduct a simple reward-based learning task. The authors even say they only recruit animals that are high food motivated. Consider removing.

A: The sentence was changed to: “The rewards were small pieces of MaplelodgeFarms® chicken hot dog” (line 204)

Line 175. This information can be added to the Table with the subject information.

A: We have added the information on the dogs’ IDs in the main text (line 206).

Line 198. Can the authors be more concrete? Why was this dog excessive while all others were not, and who was this subject? If the authors what to highlight subsets of animals, then let the reader know which ones.

A: We have added information on the dog’s ID (line 233). Since the dog was exhibiting stress constantly (barking, hiding under the chair), the session was interrupted due to welfare considerations. However, since we have counted the relative time (% of time bins, which was justified also by comparing the Learning phase with the Disruption phase, which had different durations) we did include the data of this subject in the analysis.

Line 198. “by body language and vocalizations”. Consider adding “the” before “body language”.

A: The word was added (line 234)

Lines 199-203. Despite saying this phase was similar to the Learning Phase it is unclear if the experimenters kept doing trials for the duration of the Disruption.

A: We have rewritten the paragraph so it is easier to follow and not miss the fact, that the learning task was continued during the Disruption phase (line 227)

Line 204. This seems a side point, it is unclear what is the purpose of this. How is this a food motivation control for the disruption?

A: We have added a statement on our reasoning for providing this additional phase: “The purpose of this phase was to ensure that the dogs' engagement (or lack of engagement) in the task was not predominantly impacted by satiety.” (line 239)

Line 209. How were these metrics extracted, live by the helper, or from coding recorded videos.

A: The data were extracted through the analysis of the videos.

Line 211. As stated before, dog IDs would be valuable information for the more curious reader.

A: We have provided the dogs’ IDs. (line 245).

Line 213. Who is the Experimenter 1, before authors refer to experimenter and helper. Are we missing something? Please be consistent with the terminology used.

A: We are sorry for this unintentionally inconsistent terminology. We decided to keep the terminology from the text (“the handler” and “the experimenter”) and change it in appropriate tables. As the control condition did not involve the presence of a handler (though the same team members were present in the room) we added an explanation when describing the task setup: “The owner and the handler (or one of the experimenters from the control condition, called also a handler during the task for the purpose of clarity) sat in the corner of the room.” (line 179)

Line 216. “for this phase, videos were coded for affiliative behaviours”. For the sake of comparison, and since we believe the video data are available, why not do this for the previous learning phase as well.

A: We have analysed the Learning phase videos and coded them for stress-related and affiliative behaviours, with the exception of gazing at the experimenter (see line 250). The results are presented in the text (line 318) and in Table 5.

Line 218. “The coding was conducted using a partial-interval procedure with 5 s time bins. Occurrence or non-occurrence in each 5 s time bin was noted for each behaviour”. Can the authors justify this approach? Even if the authors end up binning their metrics, it is unclear why downsampling it from the start is the best way to proceed. Also, why did the authors not consider using the duration of these events instead.

A: Although pinpoint sampling method outperforms one-zero sampling (i.e., partial interval coding), these differences are negligible given a 5 sec interval time. Nevertheless, we now recognize that, as behaviour analysts, we have “laboratory lore” that encourages the use of partial interval coding methods as tradition. We will consider using pinpoint sampling in the future.

Line 224. As previously stated, ID of the animal is useful information.

A: We have removed the information, as the tail position could also not be analysed for instance in time bins in which he dog was sitting. We have added a comment on that in the discussion (see line 469, 472)

Line 226. “The lip-licking behaviour was counted as linked to stress only if it occurred in a time bin without food delivery”. Why is this the case, is it not possible that animals are stressed during food delivery? Please elaborate.

A: During the food delivery, as the treats were moist and assumingly highly palatable, some of the dogs licked their lips immediately after receiving a piece or seemingly anticipating getting them. While stress during food delivery is possible (also when anticipating food delivery), we assumed the interpretation of this behaviour, in this case, is vague and decided to exclude the lip-licks occurring in a time bin with food delivery from the analysis.

Line 228. “by summing up the number of stress signs in each time bin and presenting the average stress score per time bin”. Once again, the authors approach is to reduce data granularity even further without a clear justification for it. Along the same lines, what does something like “presence of stress” add to a finer measurement like “average stress score per time unit”? They are highly correlated (as can be seen from Figure 3. The latter adds no information to the former. Please consider revising the data presented in the figure as well.

A: We have removed “Presence of stress”. In Fig 4 (previously Fig 3), we present The Overall stress (thus the result of our holistic scale, see below) and the Average stress score/time bin. Though they are correlated, the correlation is moderate and it is one of the problems we have encountered during the study, which we mention in the discussion section (line 460). We have used average stress score/time bin instead of summarizing the stress scores due to different duration of the phases (dog E16; differences in the duration of the Learning phase between dogs).

Line 231. Table 1. Once again, the use of experimenter 1 and 2 is not consistent, early on the manuscript they were referred to as helper and experimenter. “Proximity to the car”. The definition is not clear, plus a measuring tape was never mentioned before (or after). More generally, if these descriptions were used before, a citation would be justified, if they have not, some of the description of the target behaviours should be expanded.

A: We have carefully checked the consistency of the terminology (experimenter/handler). The ethogram was developed for this study based on the pilot trials. We have added the information on the perimeter measuring 30cm around the tub, which we missed in the text (line 187) and which is now presented also in Fig 3.

Line 237. What exactly is the statistic used to assess Inter-observer reliability? Please elaborate.

A: The statistics is expressed as percentage of agreement. We elaborated it in the paragraph on statistical analysis (line 305).

Line 242. “over-interpreted as indicative of stress”. This holistic analysis can also do exactly the opposite. All definitions (Table 2) seem very post-hoc, and the lack of granularity in the measurement alone can explain the high degree of inter-observer agreement. It is unclear what this adds the analyses and interpretation of the results.

A: We added the scale as an adjunctive measurement as it allowed us to evaluate the behaviour more holistically (as compared to coding of the body language, which for instance did not include barking or the dog’s behaviour toward the owner). We based this decision on the literature discussing different approaches

---

## [Decision Letter · Decision Letter 1]

PONE-D-24-09989R1

Exploring the impact of a brief positive experience on dogs' performance and stress resilience during a learning task

PLOS ONE

Dear Dr. Miller,

Thank you for submitting your manuscript to PLOS ONE. After careful consideration, we feel that it has merit but does not fully meet PLOS ONE’s publication criteria as it currently stands. Therefore, we invite you to submit a revised version of the manuscript that addresses the points raised during the review process.

Please see the specific reviewer comments below.

We look forward to receiving your revised manuscript.

Kind regards,

I Anna S Olsson, Ph.D.

Academic Editor

PLOS ONE

Journal Requirements:

Reviewers' comments:

Reviewer's Responses to Questions

**Comments to the Author**

1. If the authors have adequately addressed your comments raised in a previous round of review and you feel that this manuscript is now acceptable for publication, you may indicate that here to bypass the “Comments to the Author” section, enter your conflict of interest statement in the “Confidential to Editor” section, and submit your "Accept" recommendation.

Reviewer #1: (No Response)

2. Is the manuscript technically sound, and do the data support the conclusions?

Reviewer #1: Yes

3. Has the statistical analysis been performed appropriately and rigorously? 

Reviewer #1: Yes

4. Have the authors made all data underlying the findings in their manuscript fully available?

Reviewer #1: Yes

5. Is the manuscript presented in an intelligible fashion and written in standard English?

Reviewer #1: Yes

6. Review Comments to the Author

Reviewer #1: We thank the authors for their replies and changes to the manuscript, we believe it is greatly improved.

We list a few remaining points below:

Line 50. dog -> dogs

Line 93. consider rephrasing

Line 100. Please define/elaborate on the concept of "optimism" used in these experiments. The topic comes a bit out of nowhere, and a reader not familiar with this procedure is not provided with the appropriate information to deal with it.

Line 132. group -> groups

Figure 3. consider adding letters to refer to left-right panels.

Line 159. Missing link to the previous statement.

Line 169. When did the owners fill out the CBAR-Q in the experimental group? Before the walk?

Line 176. Consider rephrasing.

Line 179. Consider rephrasing, the clarification in parentheses makes the sentence more confusing.

Line 229. This simultaneously seem hard to do, given the size of the tub and the reasoning behind it is not clear…

Line 254. Unclear what it is meant here, consider rephrasing.

Line 266. This is a limitation of the study: It is not always straightforward to put a hard limit on when a behavior ends and another begins, which poses an issue when the variable is the number of such behaviors. In our view, this is the worst of both worlds: since it combines the disadvantages of binning the durations and treating behaviors like discrete units

Table 2. the arm length -> arm’s length; closer away?

Line 272. a second observer.

Line 275. Please elaborate.

Line 288. there WAS a relationship.

Line 305. This doesn't account for coincidences that may happen due to chance (as opposed to the Cohen's kappa statistic).

Results. Given that now statistical information is presented as tables, presenting p values within the test might not be necessary. That said, if the authors prefer to do so, more information should be added regarding the tests a particular sentence refers to (e.g. t value or range of values, p or largest p)

Line 320. No differences were significant.

Line 324. Full stop missing before See Table 5.

Line 326. Double full stop.

Line 341. Remove paragraph.

Line 378. Isn’t this a case to consider correcting for multiple comparisons? Or use a more sophisticated (e.g., GLM) model?

Line 384. Double full stop.

Line 391. Optimism and pessimism are not clearly defined (see previous comment for line 100).

Line 397. This is an odd sentence in the context of a scientific paper, consider removing.

Line 415. More details might be helpful to make the sheep study clearer.

Line 416. IN the Learning phase.

Line 418. “not in the laboratory”. Is this really relevant? Or rather, is it actually true to any degree? From the moment an experiment is conducted in a room, it can be considered a laboratory, sensu lato. Maybe the same holds true for the studies cited above.

Line 434. What do you been by intriguing?

Line 461. But this other measure was only used for the distraction phase, which severely limits its usefulness.

Line 473 onwards. Novel unbiased methods for automatic tracking and scoring animals’ behavior could be also mentioned in this section.

Line 484. Consider rewording.

Line 496. Why is this the case?

Line 500. But you did record CBAR-Q scores, what are we missing here?

7. PLOS authors have the option to publish the peer review history of their article (what does this mean? ). If published, this will include your full peer review and any attached files.

**Do you want your identity to be public for this peer review?** For information about this choice, including consent withdrawal, please see our Privacy Policy .

Reviewer #1: No

---

## [Author Response · Author response to Decision Letter 2]

20 Mar 2025

Reviewer #1: We thank the authors for their replies and changes to the manuscript, we believe it is greatly improved.

A: On behalf of all co-authors, I would like to thank the reviewers for dedicating their time to revise our manuscript and providing another constructive comments. Please find our detailed responses below:

Line 50. dog -> dogs

A: edited.

Line 93. consider rephrasing

A: Rephrased to: “Whereas administering pain, restraint, or isolation is easy to conceptualize as “negative” for an animal and these experiences are often included in studies on animal welfare, procedures that might induce a positive state in an animal are studied much less frequently”

Line 100. Please define/elaborate on the concept of "optimism" used in these experiments. The topic comes a bit out of nowhere, and a reader not familiar with this procedure is not provided with the appropriate information to deal with it.

A: We have removed this section from the initial version in an attempt to shorten the introduction. However, we agree that it now may be unclear to a reader, who is not familiar with JBT. We have added an elaboration (line 99)

Line 132. group -> groups

A: edited.

Figure 3. consider adding letters to refer to left-right panels.

A: We have added letters to the figure.

Line 159. Missing link to the previous statement.

A: rephrased to: “The activities were chosen, based on previous research that found that human social interaction, access to toys, and olfactory search induce ‘optimism’ in dogs [53,54]. To encourage olfactory search, and given that dogs benefit from food-stuffed toys we added treats as one of the elements of the positive experience session [59]” line 161

Line 169. When did the owners fill out the CBAR-Q in the experimental group? Before the walk?

A: We have added an explanation (line 176). Basically, we held a general conversation with the owners and also explained to them how to fill out the questionnaire. Depending on the number of questions they had, some owners managed to do it during this session, and some had to do it (or finish it) after the study. The owners of the dogs from the experimental the dogs filled out the questionnaire after the study. That is why, if the dog did not reach the learning criteria and had to be excluded from the study, we have not obtained the C-BARQ scores.

Line 176. Consider rephrasing.

A: we have edited the sentence, now line 183

Line 179. Consider rephrasing, the clarification in parentheses makes the sentence more confusing.

A: we wanted to avoid double nomenclature, that was present in the initial submission. We shortened the sentence in the parenthesis (line 186)

Line 229. This simultaneously seem hard to do, given the size of the tub and the reasoning behind it is not clear…

A: The car is small and moves easily in four directions. During the pilot studies, we have seen big differences between the sounds that can be produced by the car depending on how we operate and who is operating the car. Eventually, we have developed a unified way of moving the car back and forth and to the sides, that was possible to be kept consistent between dogs.

Line 254. Unclear what it is meant here, consider rephrasing.

A: Rephrased to: “Note that affiliative behaviours toward the experimenter were not coded in the Learning phase because it was impossible to distinguish the dog’s motivation e.g. for gazing at the experimenter at the time when they did not know the task and were engaged in learning” line 261

Line 266. This is a limitation of the study: It is not always straightforward to put a hard limit on when a behavior ends and another begins, which poses an issue when the variable is the number of such behaviors. In our view, this is the worst of both worlds: since it combines the disadvantages of binning the durations and treating behaviors like discrete units

A: We thank the reviewer for this observation. However, we believe that our text was misleading. We have now amended the text to indicate that each time bin was scored in a binary fashion (behaviour present or absent) not as counts of frequency of behaviours within that time bin. The reason we present an average stress score/time bin is the fact, that the length of the Learning phase was not consistent across dogs, and also the Disruption phase was shorter for a dog expressing excessive stress sings. This method, a partial interval coding method or one-zero sampling, has been previously used in animal behavioural studies and has the advantage of standardizing behaviours that are discreet to behaviours that are continuous. Every recording method has its advantages and disadvantages, but we believe this method allows for sufficient precision (with our length of time bins) as well as allows for comparisons across behaviours.

Table 2. the arm length -> arm’s length; closer away?

A: we have edited it in the table.

Line 272. a second observer.

A: edited

Line 275. Please elaborate.

A: We have elaborated it to: ”The highest discrepancies were seen in coding “tail down” (58.3% - 100%), therefore this behaviour was double-coded in all of the videos from the Disruption phase with a mean agreement of 96.2% ± 8.1%”. However, it is hard to say why we have seen these interpretation problems, we can only assume it is because there were some differences in interpreting the low tail position (hanging tail vs tail forcefully held down). This was one of the reasons we added the holistic scale. line 283

Line 288. there WAS a relationship.

A: edited

Line 305. This doesn't account for coincidences that may happen due to chance (as opposed to the Cohen's kappa statistic).

A: we agree. We used Cohen’s kappa for the overall stress score. For single behaviours (24 time bins for each dog, 24 different behaviours, of which some were summarized to stress score) we have decided to keep the percentages of agreement though it is for sure a less robust parameter.

Results. Given that now statistical information is presented as tables, presenting p values within the test might not be necessary. That said, if the authors prefer to do so, more information should be added regarding the tests a particular sentence refers to (e.g. t value or range of values, p or largest p)

A: we have removed the p-values for parameters, that are presented in the tables.

Line 320. No differences were significant.

A: edited

Line 324. Full stop missing before See Table 5.

A: edited

Line 326. Double full stop.

A: edited

Line 341. Remove paragraph.

A: edited

Line 378. Isn’t this a case to consider correcting for multiple comparisons? Or use a more sophisticated (e.g., GLM) model?

A: Our study set out to analyse the effects of a positive experience (pre-treatment) on subsequent learning performance (evaluated in the Learning phase) and susceptibility to stress in the presence of a disruptor (evaluated in the Disruption phase).

As such, we explored two different hypotheses.

• Effect of pre-treatment on performance (experimental vs control in the learning phase).

• Effect of pre-treatment on stress resilience (experimental vs control in the learning phase)

These analyses were run separately, accounting for parameters which were present on only one of the phases (e.g., behaviours related to the stressor or latency to reach the learning criterion).

We also included other parameters (like stress coding in the Learning phase) to answer some questions that arose in the review process, such as whether the remotely controlled car could be considered a stressor in the first place (for this purpose, we have used Wilcoxon rank-signed test).

We will consider a GLM approach for our next manuscript in which more groups and phases are involved.

Line 384. Double full stop.

A: edited

Line 391. Optimism and pessimism are not clearly defined (see previous comment for line 100).

A: we elaborated the definition of optimism as defined in JBT in the introduction

Line 397. This is an odd sentence in the context of a scientific paper, consider removing.

A: We removed the sentence

Line 415. More details might be helpful to make the sheep study clearer.

A: We elaborated this part of the discussion (see line 418)

Line 416. IN the Learning phase.

A: edited

Line 418. “not in the laboratory”. Is this really relevant? Or rather, is it actually true to any degree? From the moment an experiment is conducted in a room, it can be considered a laboratory, sensu lato. Maybe the same holds true for the studies cited above.

A: yes, we agree. Our point was to explain that it was not conducted in the same room/space as the learning and disruption phases. But we agree that it might not be seen as that much different by the dogs, this is why consider prolonged habituation as one of the explanations.

Line 434. What do you been by intriguing?

A: We have toned it down to: “interesting”

Line 461. But this other measure was only used for the distraction phase, which severely limits its usefulness.

A: After the first review we have attempted to recode the learning phase as well, however, we decided that this is not a useful comparison given the lack of the car, which alters the operational definition of that scale. The limitations of the holistic scale are discussed in the discussion section.

Line 473 onwards. Novel unbiased methods for automatic tracking and scoring animals’ behavior could be also mentioned in this section.

A: We have added references on this topic, see line 492, references 77-79.

Line 484. Consider rewording.

A: The sentence was shortened (line 496)

Line 496. Why is this the case?

AND

Line 500. But you did record CBAR-Q scores, what are we missing here?

A: We have explained it by: “Another limitation is the lack of detailed questionnaires from the owners of some dogs who did not reach the learning criteria. For these dogs, we did not collect C-BARQ scores after the experiment if the owner did not fill out the questionnaire during the experiment (e.g. due to limited time).” Line 510

---

## [Editor Report · Decision Letter 2]

Exploring the impact of a brief positive experience on dogs' performance and stress resilience during a learning task

PONE-D-24-09989R2

Dear Dr. Miller,

We’re pleased to inform you that your manuscript has been judged scientifically suitable for publication and will be formally accepted for publication once it meets all outstanding technical requirements.

Kind regards,

I Anna S Olsson, Ph.D.

Academic Editor

PLOS ONE
---

## [Editor Report · Acceptance letter]

PONE-D-24-09989R2

PLOS ONE

Dear Dr. Miller,

I'm pleased to inform you that your manuscript has been deemed suitable for publication in PLOS ONE. Congratulations! Your manuscript is now being handed over to our production team.

Kind regards,

on behalf of

Dr. I Anna S Olsson

Academic Editor

PLOS ONE